EMBO
Molecular Medicine

# TREM2 shedding by cleavage at the H157-S158 bond is accelerated for the Alzheimer's disease-associated H157Y variant

Peter Thornton[1,†], Jean Sevalle[2,†], Michael J Deery[3,#], Graham Fraser[1], Ye Zhou[2], Sara Ståhl[4], Elske H Franssen[1], Roger B Dodd[5,6], Seema Qamar[5], Beatriz Gomez Perez-Nievas[1], Louise SC Nicol[6], Susanna Eketjäll[7], Jefferson Revell[6], Clare Jones[6], Andrew Billinton[1], Peter H St George-Hyslop[2,5,‡], Iain Chessell[1,‡] & Damian C Crowther[1,‡,*] (iD)

## Abstract

We have characterised the proteolytic cleavage events responsible for the shedding of triggering receptor expressed on myeloid cells 2 (TREM2) from primary cultures of human macrophages, murine microglia and TREM2-expressing human embryonic kidney (HEK293) cells. In all cell types, a soluble 17 kDa N-terminal cleavage fragment was shed into the conditioned media in a constitutive process that is inhibited by G1254023X and metalloprotease inhibitors and siRNA targeting ADAM10. Inhibitors of serine proteases and matrix metalloproteinases 2/9, and ADAM17 siRNA did not block TREM2 shedding. Peptidomimetic protease inhibitors highlighted a possible cleavage site, and mass spectrometry confirmed that shedding occurred predominantly at the H157-S158 peptide bond for both wild-type and H157Y human TREM2 and for the wild-type murine orthologue. Crucially, we also show that the Alzheimer's disease-associated H157Y TREM2 variant was shed more rapidly than wild type from HEK293 cells, possibly by a novel, batimastat- and ADAM10-siRNA-independent, sheddase activity. These insights offer new therapeutic targets for modulating the innate immune response in Alzheimer's and other neurological diseases.

**Keywords** genetic risk; microglia; neurodegeneration; neuroinflammation
**Subject Categories** Genetics, Gene Therapy & Genetic Disease; Immunology; Neuroscience

See also: **K Schlepckow et al** (October 2017)

## Introduction

Nasu–Hakola disease is a rare but fatal brain and bone disorder (Hakola, 1972; Hakola & Iivanainen, 1973), caused by homozygous inheritance of null or hypomorphic variants of the TREM2 gene (Klünemann et al, 2005). A subset of heterozygous TREM2 variants such as R47H, R62H and H157Y are associated with increased risk of Alzheimer's disease (AD) (Guerreiro et al, 2013; Jonsson & Stefansson, 2013; Jonsson et al, 2013; Slattery et al, 2014; Finelli et al, 2015; Ghani et al, 2016; Jiang et al, 2016). While the prevalence of AD-linked TREM2 variants is low, indeed the most common (R47H) affects 0.3–0.6% of the population (Guerreiro et al, 2013; Jonsson et al, 2013), the relative risk for individual carriers is high, twofold to 11-fold above the general population (Finelli et al, 2015).

TREM2 is a single-pass, type I transmembrane protein that includes an extracellular immunoglobulin domain with two N-linked glycans (Park et al, 2015). It is expressed on dendritic cells, macrophages, microglia and osteoclasts and is thought to act as a scavenger receptor (Colonna, 2003). TREM2 is prominent in microglia adjacent to pathological material such as amyloid deposits and cellular debris and may have a role in directing microglial migration (Jay et al, 2015; Kawabori et al, 2015) and phagocytic activation (Keren-Shaul et al, 2017). Studies in cell cultures have shown that at least Nasu–Hakola-linked TREM2 variants are less available on the surface of phagocytic cells because they are incorrectly

1  Neuroscience, Innovative Medicines and Early Development, AstraZeneca, Granta Park, Cambridge, UK
2  Tanz Centre for Research in Neurodegenerative Diseases, University of Toronto, Toronto, ON, Canada
3  Cambridge Centre for Proteomics, University of Cambridge, Cambridge, UK
4  AstraZeneca Translational Sciences Centre, Karolinska Institutet, Stockholm, Sweden
5  Department of Clinical Neurosciences, Cambridge Institute for Medical Research, University of Cambridge, Cambridge, UK
6  MedImmune Limited, Granta Park, Cambridge, UK
7  Cardiovascular and Metabolic Diseases, Innovative Medicines and Early Development, AstraZeneca, ICMC, Huddinge, Sweden
   *Corresponding author. Tel: +44 020 3749 6149; E-mail: damian.crowther@azneuro.com
   †These authors contributed equally to this work
   ‡These authors contributed equally to this work
   #Correction added on 2 October 2017 after first online publication: "Mike J Deery" was corrected to "Michael J Deery".

**A**

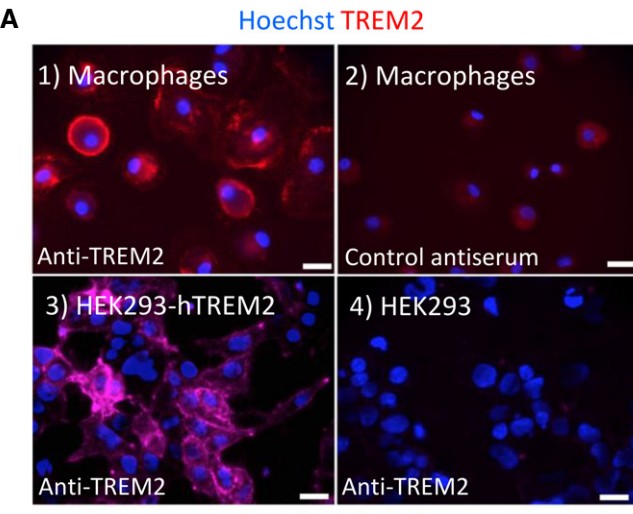

**B**

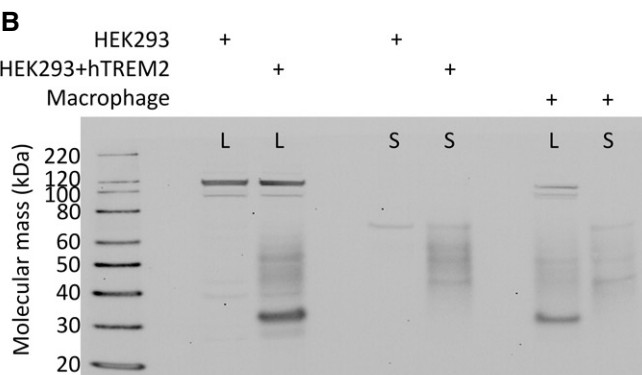

S = culture supernatant
L = cell lysate

**C**

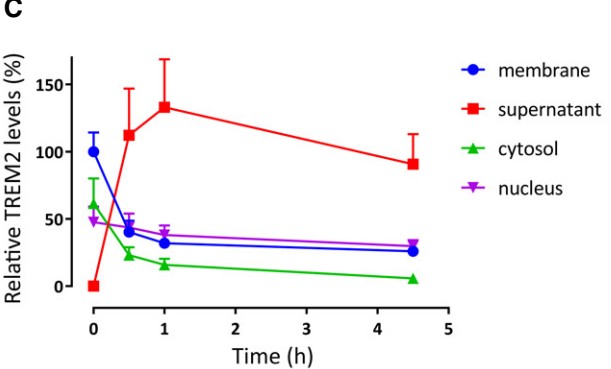

**Figure 1.   TREM2 expression, glycosylation and proteolysis.**

A   Surface TREM2 was detected on non-permeabilised primary human macrophages labelled with anti-TREM2 polyclonal antiserum (1, red), but not a control antiserum (2) and by live cell immunostaining of HEK293 stably transfected with wild-type hTREM2 (3, pink; nuclei stained with Hoechst) but not on parental HEK293 (4). Surface immunolocalisation was also observed. Scale bars = 20 μm.

B   Western blots of lysates (L) and supernatants (S) for hTREM2 from parental HEK293 vs. HEK293+hTREM2 cells showed distinct isoforms of TREM2. The cell lysate (HEK293+hTREM2, L) yielded an immature glycoform at 35 kDa with a less intense smear up to 50 kDa. This smear was the predominant species in the supernatant (HEK293+hTREM2, S). Similar distributions of TREM2 were seen in primary human macrophages (Macrophage, L and S).

C   Subcellular fractionation of macrophages over a time course revealed the fate of surface-biotinylated TREM2 (membrane-associated in blue circles), indicating that most protein was shed into the supernatant (red squares), with a half time of < 1 h, and that little was found in cytosol or nuclear cellular fractions (green triangles & purple inverted triangles). Data plotted as mean ± SEM; *n* = 4 replicates.

Source data are available online for this figure.

In this study, we have investigated the properties of the protease (the sheddase) that results in shedding of TREM2 from the surface of primary human and murine myeloid cells. We find that the sheddase activity is sensitive to metalloprotease inhibitors and cleaves at the H157-S158 peptide bond, in both primary human macrophages and mouse microglia. We investigate the effect of the disease-associated H157Y substitution and show that it may recruit a novel sheddase, possibly explaining its pathogenic mechanism.

## Results

### TREM2 is expressed on the surface of human cells and the N-terminal fragment is shed into the conditioned medium

Non-permeabilised primary human macrophages showed high levels of TREM2 expression on their cell surfaces when probed with the AF1828 polyclonal antibody (Fig 1A, 1, red) but not with control goat serum (Fig 1A, 2). Likewise, stable-expressing hTREM2-HEK293 cells showed TREM2 on their surface (Fig 1A, 3, pink), which was not seen in the untransfected parental cells (Fig 1A, 4). In both cell types, Western blots of the cell lysates and conditioned media showed a range of TREM2 isoforms (Fig 1B). The lysates (L) contained a predominant 35 kDa band with a less intense smear up to 50 kDa. By contrast, this higher molecular weight smear is the major TREM2 species in the culture supernatants (S) from both HEK293 + hTREM2 and human primary macrophages.

We investigated the kinetics of TREM2 metabolism in primary human macrophages by biotin pulse labelling of surface-exposed protein and then following the fate of biotin-conjugated TREM2 in various subcellular fractions over time (Fig 1C, raw data Fig EV1). It is notable that the surface-exposed TREM2 has a short half-life of < 1 h and that the bulk of the lost protein appears in the supernatant. Very little surface-expressed TREM2 was seen to internalise into cytoplasmic or nuclear compartments under basal conditions.

glycosylated (Kleinberger *et al*, 2014; Park *et al*, 2015) and inefficiently trafficked through the synthetic pathway. Most individuals with AD carry wild-type TREM2 and the normal proteolytic processing of the protein is thought to be performed by sequential α- then γ-secretase activity (Wunderlich *et al*, 2013). It is notable that levels of the shed TREM2 N-terminal fragment (NTF) are raised in the CSF of patients with sporadic AD as compared to healthy controls (Heslegrave *et al*, 2016; Piccio *et al*, 2016; Suárez-Calvet *et al*, 2016b).

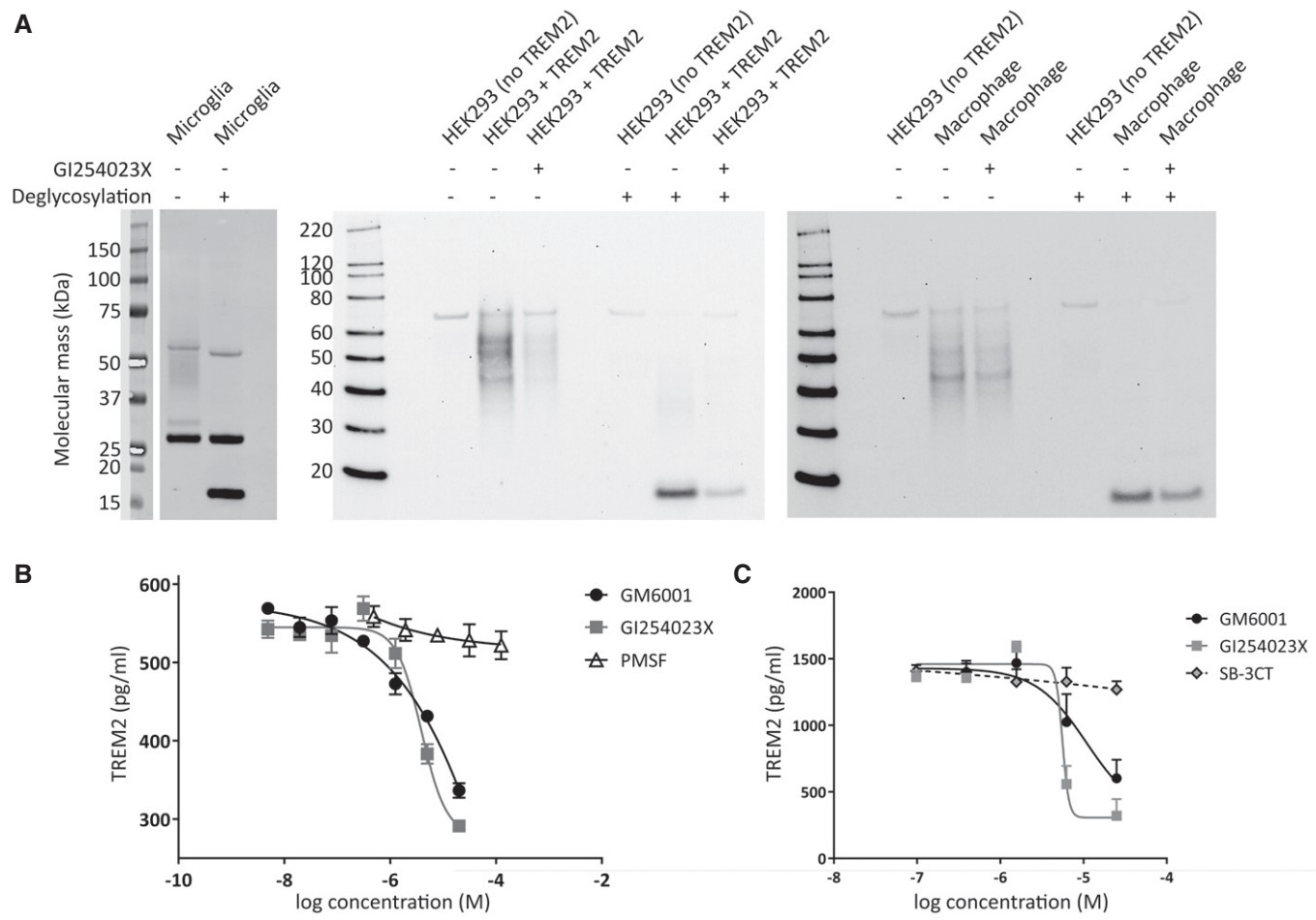

**Figure 2. Shedding of glycosylated TREM2 NTF is sensitive to inhibitors of ADAM10 and matrix metalloproteinases.**

A  Western blotting of the conditioned media from primary murine microglia (left panel), transfected HEK293 cells (middle panel) and primary human macrophages (right panel) revealed the presence of shed wild-type TREM2. This soluble TREM2 appeared as a > 35 kDa smear of various glycoforms and upon deglycosylation was reduced to a single band of 17 kDa (arrowhead). The ADAM10 inhibitor, GI254023X, blocked shedding in transfected HEK293 cell and macrophage cultures.

B  The concentration of shed TREM2 in the supernatant of macrophage cultures was reduced to ~50% by 20 μM of both GI254023X and the broad-spectrum metalloprotease inhibitor GM6001, as measured by an MSD assay. By contrast, the broad-spectrum serine protease inhibitor PMSF did not reduce shedding. Experiments were repeated three times and for two donors of the macrophage progenitors. Data plotted as mean ± SEM.

C  In HEK293 cells, GI254023X and GM6001 had comparable potencies; however, the MMP2/9 inhibitor SB-3CT did not block shedding. Data plotted as mean ± SEM; $n = 3$ replicates.

Source data are available online for this figure.

## Proteolytic release of a 17 kDa TREM2 N-terminal fragment into the supernatants of primary mouse microglia, primary human macrophages and HEK293 cells

Soluble TREM2 NTF was detected by Western blotting the conditioned medium from primary murine microglia and primary human macrophages; a similar result was also seen for HEK293 cells (Fig 2A). The high molecular mass smear of immunoreactivity (> 35 kDa) was likely due to extensive and variable glycosylation of the NTF because only a single band of 17 kDa was observed following deglycosylation. In primary human macrophages, and in wild-type hTREM2-expressing HEK293 cells, the ADAM10-specific metalloprotease inhibitor GI254023X reduced shedding. The potencies of GI254023X and a broad-spectrum metalloprotease inhibitor (GM6001) were quantified by a Meso Scale Discovery (MSD) assay for shed TREM2 and were

shown to be similar in macrophages (Fig 2B), inhibiting shedding by ~50% at 20 μM. Under the same conditions, the broad-spectrum serine protease inhibitor PMSF did not inhibit TREM2 shedding at concentrations exceeding 100 μM (Fig 2B). GI254023X and GM6001 also inhibited shedding in HEK293 cells; additionally, the MMP2/9 inhibitor SB-3CT failed to inhibit shedding (Fig 2C).

## Peptidomimetic protease inhibitors point to residues 158–160 as the site of sheddase cleavage

We synthesised a tiled library of reverse-sequence D-amino acid (retro-inverso) polypeptides that replicated both the biophysical characteristics and the specific side-chain interactions of peptides constituting the peri-membrane region of TREM2 (Li *et al*, 2010) (Fig 3A). Unlike natural L-amino acid polypeptides, these

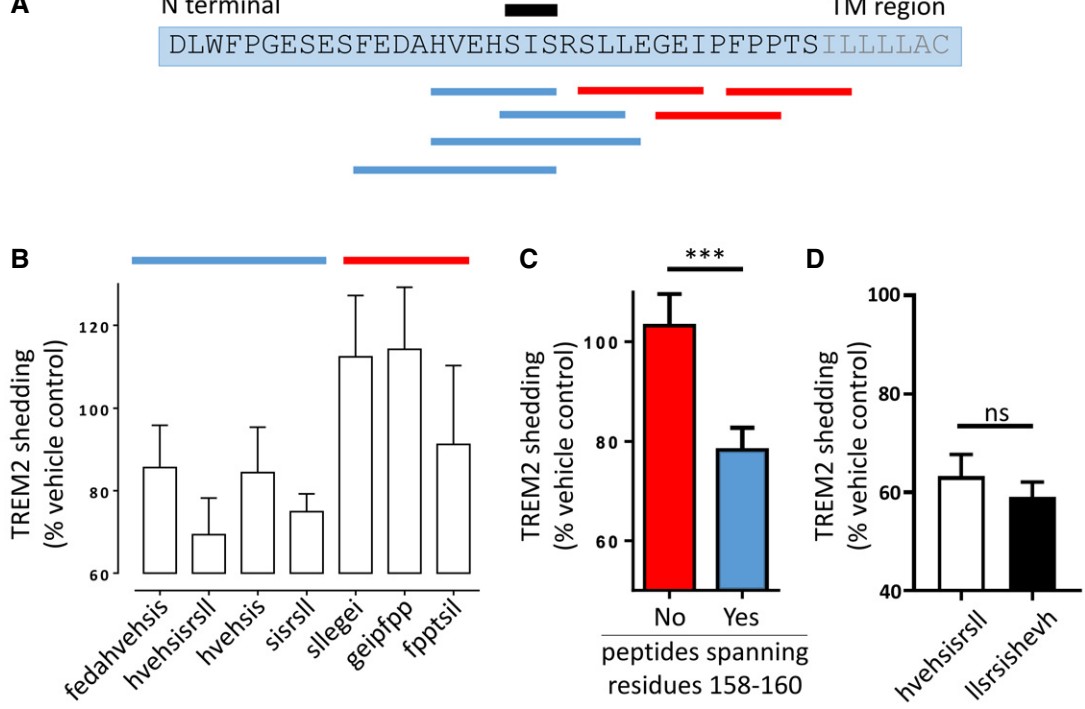

**Figure 3.  Peptidomimetic protease inhibitors point to residues 158–160 as the site of sheddase cleavage.**

A  An overlapping library of retro-inverso peptides were designed to mimic the extracellular peri-membranous domain of TREM2 in the region of the sheddase site (TM: transmembrane). Blue boxes represent retro-inverso peptides that reduce TREM2 shedding; red boxes those that do not. The black box indicates the three residues that are common to all the inhibitory retro-inverso peptides.

B  The peptidomimetics were incubated with primary human macrophages, and the resulting levels of shed TREM2 NTF were quantified by MSD ELISA. Blue = inhibitory; red = non-inhibitory.  Values plotted: mean ± SEM; each experiment was repeated for 3–5 independent human donors.

C  Peptides including amino acids 158–160 (blue) inhibited TREM2 shedding more than retro-inverso peptides that did not (red). Values plotted: mean ± SEM; two-tailed Student's *t*-test, ***P = 0.003; each experiment was repeated for 3–5 independent human donors.

D  Forward and reverse TREM2 peptidomimetics containing residues 158–160 suppressed TREM2 shedding equally. Values plotted: mean ± SEM; *n* = 3 replicates; two-tailed Student's *t*-test; ns = not significant.

peptidomimetics are resistant to proteolysis (Taylor *et al*, 2010) and at millimolar concentrations predictably act as competitive protease inhibitors. The levels of TREM2 NTF in the conditioned media of macrophage cultures treated with the peptidomimetics were compared with the levels in the media of untreated cultures. We found that all retro-inverso peptides that included residues 158–160 (C'-fedahvehsis-N', C'-hvehsisrsll-N', C'-hvehsis-N' & C'-sisrsll-N') inhibited shedding, whereas other members of the library (C'-sllegei-N', C'-geipfpp-N' & C'-fpptsil-N'), analogous to nearby peri-membrane regions of TREM2, did not (Fig 3B and C).

To understand whether the protease inhibition was strictly sequence specific, we synthesised a D-polypeptide with the reverse sequence of the most effective inhibitor (C'-hvehsisrsll-N'). This reverse retro-inverso peptide was equally effective at preventing TREM2 shedding (Fig 3D), indicating that access to the protease is determined less by the specific amino acid sequence as by general biophysical characteristics such as charge.

**Mass spectrometry identifies His157-Ser158 as the main sheddase cleavage site in wild-type and disease-linked variants of TREM2**

Immunoprecipitation, using a goat polyclonal anti-hTREM2 (AF1828, R&D Systems) from the conditioned medium of primary

human macrophages, followed by deglycosylation, yielded two specific bands on a silver-stained protein gel (Fig 4A, single biological replicate run in duplicate lanes). The upper band had a molecular mass of 17 kDa, consistent with the expected size of the NTF (arrowhead 1); running slightly ahead of this was a 15 kDa protein (arrowhead 2). SDS–PAGE purification and trypsin digestion of these bands yielded peptides that were analysed by LC-MS/MS, using the published TREM2 sequence as the search guide (PubMed references: human: NP_061838.1 & mouse: NP_001259007.1). Similar mass spectrometric assays were performed independently and in duplicate for the conditioned media of primary human macrophages, primary murine microglia and HEK293 cells stably expressing hTREM2. Murine TREM2 was immunoprecipitated with a rat anti-mTREM2 monoclonal (MAB1729, R&D Systems). The conditioned media were derived from two donors (macrophages and microglia) or from two independent cultures of HEK293 cells. Those peptides with C-terminal Arg and Lys residues were considered to result from trypsin cleavage while peptides with other C-terminal residues were likely the result of sheddase activity. For experiments that yielded non-trypsin-derived peptide fragments, we plotted their frequency against the peri-membrane sequence of TREM2 (Fig 4B).

Mass spectrometry yielded almost complete coverage of the extracellular domain of TREM2 (Fig 4C). For the 17 kDa band 1

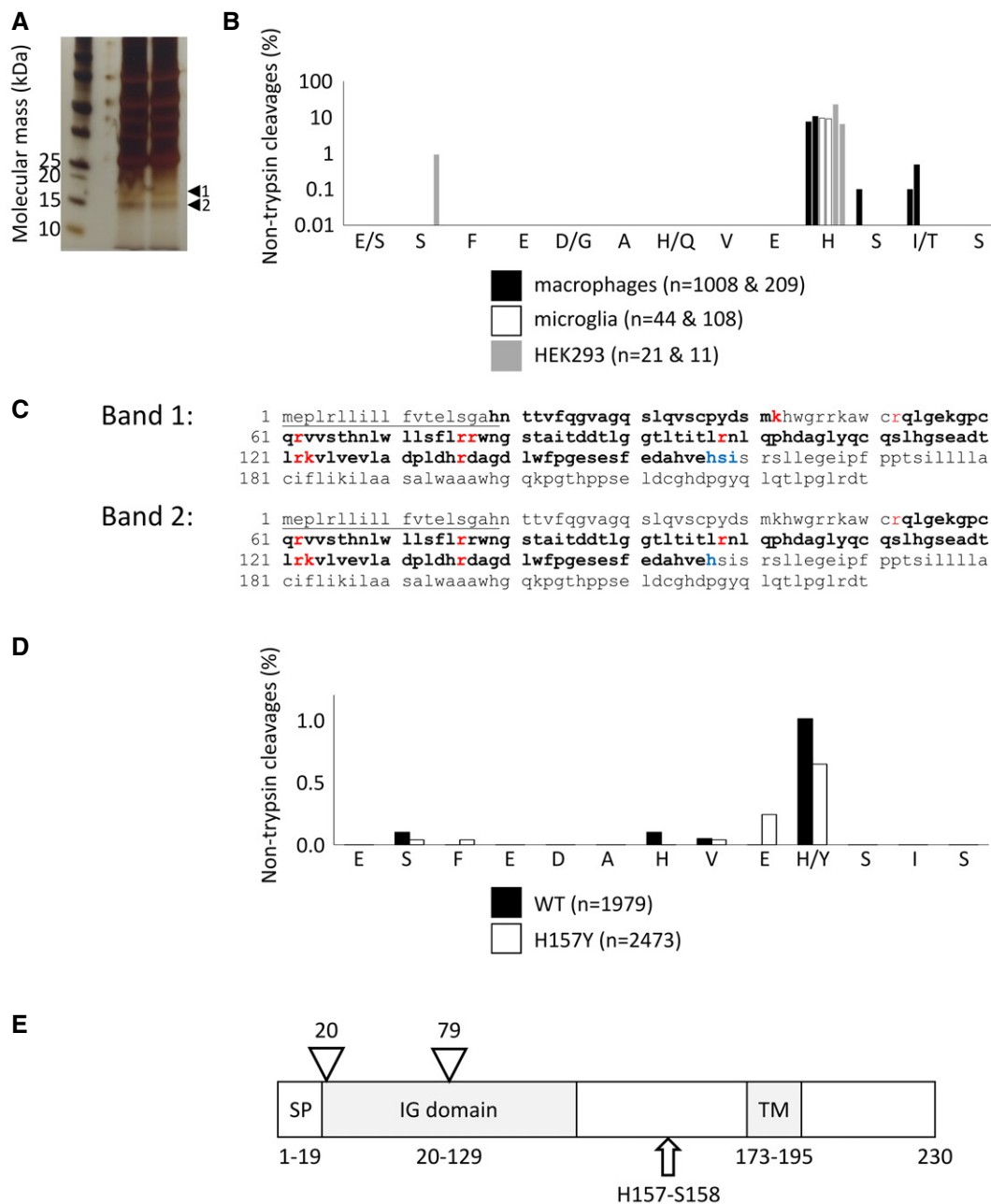

**Figure 4. Mass spectrometry identifies His157-Ser158 as the sheddase site.**

A   TREM2 was immunoprecipitated and deglycosylated from primary human macrophage conditioned media. Two specific bands were visible on a silver-stained SDS–PAGE gel (arrowheads 1 & 2). These bands were excised, digested with trypsin and analysed by LC-MS/MS.

B   TREM2 in the conditioned media of primary human macrophages (black), primary murine microglia (white) and HEK293 cells stably expressing hTREM2 (grey) was digested with trypsin and the resulting peptides identified by mass spectrometry. The most frequent C-terminal residue not consistent with trypsin digestion was H157 (from two donors/biological replicates of HEK293 cells; each replicate assayed in separate mass spectrometry laboratories; n = total number of peptides identified; where peptide sequences differ between species they are shown as human/mouse).

C   Trypsin digestion (red sites) of band 1 provided almost complete coverage of macrophage TREM2 (bold), lacking only the peptide expected to have R52 at its C-terminus. Non-trypsin cleavage (blue) was observed predominantly at H157. The absence of the peptide with K42 at its C-terminus suggests that N-terminal truncation is responsible for generating band 2. Underlined: predicted secretion signal peptide.

D   Peptides from the supernatants of HEK293 cells transiently expressing wild-type (black) and H157Y (white) human TREM2, and co-expressing human DAP12, were identified by mass spectrometry. The most common C-terminus was residue 157 for both TREM2 isoforms (one biological replicate; n = total number of peptides identified; where WT and variant sequences vary they are shown as WT/variant).

E   Schematic of the TREM2 protein. SP, signal peptide; IG domain, immunoglobulin domain; TM, transmembrane domain; triangles, *N*-glycosylation sites; arrow, site of proteolytic shedding; all numbers relate to amino acid positions.

Source data are available online for this figure.

   

from primary human macrophages, we observed peptides that included the N-terminus through to the most distal non-trypsin cleavage sites. These cleavages, likely due to the sheddase, occurred most commonly at H157, with such peptides accounting for up to 10% of all TREM2 peptides detected. Other nearby non-trypsin sites at positions 158 and 159 each accounted for < 0.1% of the peptides observed. Across all three cell types, H157 was the most common site with other non-trypsin sites each accounting for < 1% of the total peptides observed. The 15 kDa band 2, from primary human macrophages, yielded similar peptide fragments but lacked the most N-terminal sequence, making it likely that a second endoproteolytic event within the first 34 amino acids of TREM2 is a common occurrence. Control experiments in which cell-free, full-length, recombinant TREM2 was digested with trypsin did not yield peptides with H157 at the C-terminus, instead the expected tryptic peptides ending with R161 were observed (data not shown).

In the same way, we identified peptides in the conditioned medium of HEK293 cell lines transiently expressing wild-type (Fig 4D, black) and H157Y (white) human TREM2 and co-expressing human DAP12. For both isoforms, residue 157 was the predominant C-terminal residue indicating that the site of proteolysis is largely unaffected by the substitution of a non-polar tyrosine for the polar histidine at the P1 position (see Fig 4E for schematic of cleavage site).

## The H157Y substitution accelerates TREM2 shedding from HEK293 cells

The H157Y variant of TREM2 carries an increased risk of AD (Jiang *et al*, 2016) although, like the R47H variant, the mechanism is unknown. To test whether H157Y substitution alters the proteolytic metabolism of TREM2, we expressed the wild-type and the variant proteins in HEK293 cells. Levels of cell-associated TREM2 in cell lysates (Fig 5A) and shed TREM2 in conditioned media (Fig 5C) TREM2 were measured by Western blot (full-length TREM2 species quantified in Fig 5B and cleaved fragments in Fig 5D). Both wild-type and H157Y variant TREM2 expressed at similar levels. However, for H157Y TREM2 there was a reduction in mature glycoforms in the cell lysates, whereas the

immature glycoforms were equivalent (blot in Fig 5A, quantified in Fig 5B). This altered ratio is unlikely the result of impaired delivery of the variant to the cell surface because levels of both the TREM2 C-terminal fragment (CTF, blot in Fig 5A, quantified in Fig 5D) and NTF (blot in Fig 5C, quantified in Fig 5D) are higher for the H157Y variant as compared to wild type. Rather it appears that mature H157Y TREM2 is shed more rapidly from the cell surface.

We then treated the HEK293 cells expressing TREM2 with protease inhibitors GI254023X and batimastat. Remarkably, while GI254023X inhibited the shedding of wild-type and H157Y variant TREM2 equally (Fig 5E), by contrast batimastat was only partially effective in preventing the shedding of H157Y variant TREM2 (Fig 5F; batimastat titration in Fig EV2; Fig 5G: schematic of expected protease activities). This may indicate that the H157Y variant is susceptible to a novel proteolytic activity. We repeated these assays of TREM2 proteolysis in HEK293 cells expressing the equivalent halotagged constructs with essentially identical results (Fig EV3).

## Knock-down of ADAM10 is particularly effective at reducing the shedding of wild-type TREM2

HEK293 cells expressing either WT or H157Y TREM2 were treated with pooled siRNA constructs (Dharmacon) targeting either ADAM10 and/or ADAM17 (Fig 6A and B). After incubating the treated cells for 72 h, the concentration of TREM2 NTF in the conditioned media was measured by MSD assay. ADAM10, but not ADAM17, knock-down reduced the shedding of TREM2 NTF for both WT and the H157Y variant. Control experiments indicated that the levels of TREM2 in the cell pellet were unchanged by siRNA treatment ($N = 2$, data not shown). The blockade of TREM2 shedding, as measured by reduction in the levels of the NTF in the conditioned medium following ADAM10 siRNA, was greater for the WT protein (45.0%) as compared to the variant (30.0%, $N = 8$, $P = 10^{-4}$, one-tailed Student's *t*-test, Fig 6C). This result further supports the involvement of proteases other than ADAM10, but not ADAM17, in the shedding of the H157Y variant.

**Figure 5. The disease-linked H157Y variant of TREM2 is shed more rapidly than wild-type TREM2.**

A   Western blot for TREM2 in lysates of HEK293 cells transiently expressing either wild-type (WT) or the H157Y variant protein ($N = 3$): levels of immature TREM2 (major band at 35 kDa) were unchanged by the H157Y substitution; however, total levels of the variant were reduced as compared to WT because of a more marked reduction in the levels of the glycosylated isoform. The proteolytic cleavage of TREM2 generated a truncated C-terminal fragment (CTF) that was more abundant in lysates from cells expressing H157Y TREM2. GAPDH was the loading control; DAP12 was co-expressed with TREM2. Molecular mass markers in kDa.

B   Quantitation of the full-length TREM2 isoforms as shown in panel (A) (data plotted as mean ± SEM; $N = 12$).

C   Western blot for the shed TREM2 NTF from the conditioned medium of HEK293 cell cultures ($N = 3$): levels of H157Y TREM2 NTF were higher than WT. A secreted fragment of the amyloid precursor protein (sAPPα) was the loading control. Molecular mass markers in kDa.

D   The proteolytic fragments of TREM2 as shown in panel (A) (CTF, $N = 3$) and panel (C) (NTF, $N = 15$) were corrected for the total full-length TREM2 (FL) from each cell lysate: the levels of the shed N-terminal fragment (NTF) of TREM2 were higher in cells expressing the H157Y variant as compared to WT. Data plotted as mean ± SEM.

E   Western blot for TREM2 from the conditioned medium of HEK293 cell cultures treated with varying concentrations of either GI254023X or batimastat (bat): inhibition of TREM2 NTF shedding by GI254023X was equivalent for both variant and WT TREM2; however, more shedding was observed at the higher concentration of batimastat for H157Y TREM2 as compared to WT. Molecular mass markers in kDa.

F   Quantification of total TREM2 shed from HEK293 cells as shown in panel (E) ($N = 7$). Maximal concentrations of GI254023X blocked shedding equally for WT and variant TREM2. Batimastat-resistant shedding was more marked for H157Y TREM2. Data plotted as mean ± SEM.

G   Schematic showing the proteolytic enzymes expected to be active in the presence of the protease inhibitors used in (E and F). MMP: matrix metalloproteinase.

Data information: Concentration of inhibitors in micromolar. Two-tailed Student's *t*-test, *P*-values: **P* < 0.05; ****P* < 0.001.

Source data are available online for this figure.

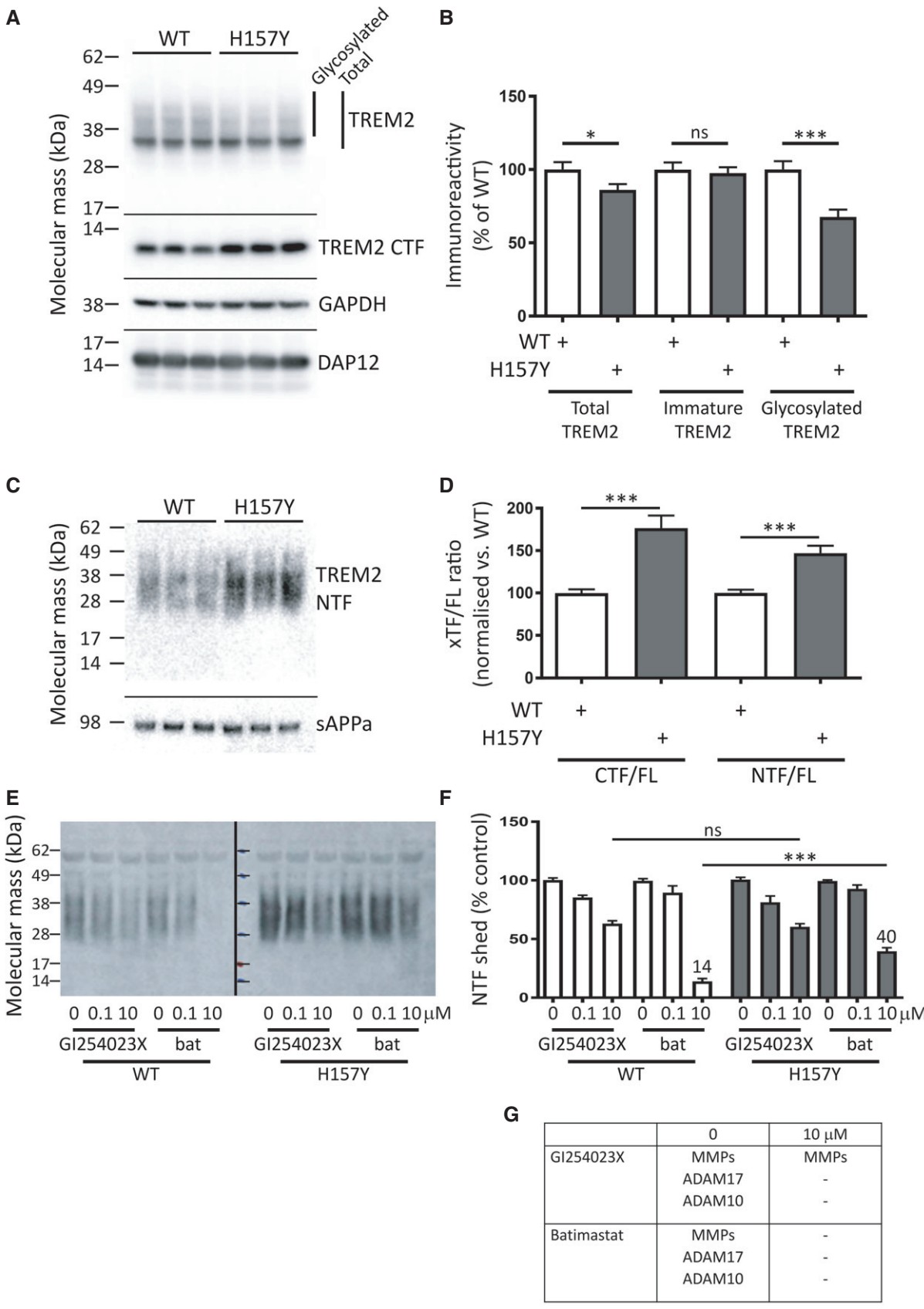

**Figure 5.**

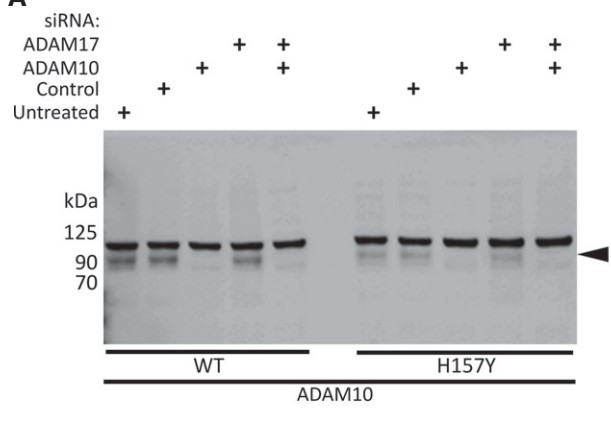

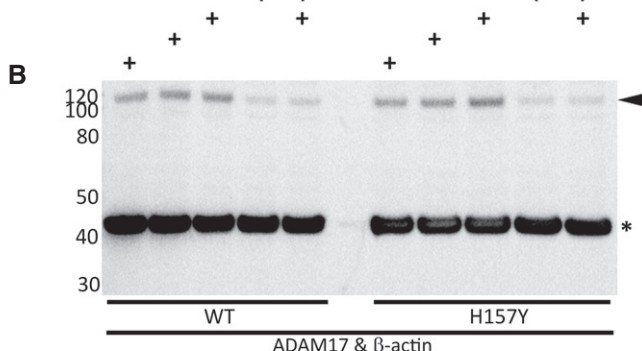

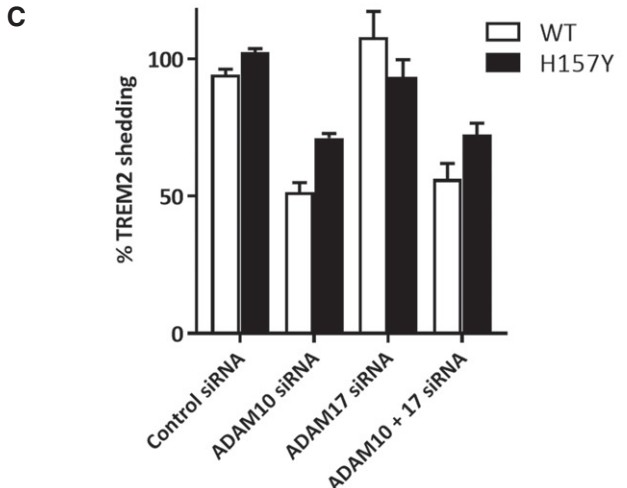

**Figure 6. Knock-down of ADAM10 is less effective at reducing the shedding of H157Y TREM2.**

A, B  Western blot of HEK293 cells for ADAM10 (A) or ADAM17 and reprobed for β-actin (B) confirmed siRNA-mediated knock-down of target proteins (arrowheads). siRNA pools, or vehicle, added to cells as indicated. β-actin (asterisk) was the loading control.

C  Quantitation of WT and H157Y (SEM error bars) TREM2 NTF in the conditioned media of HEK293 cells as compared to untreated cells measured by MSD assay (N = 8): ADAM10 siRNA reduced shedding whether applied alone or in combination with ADAM17 siRNA. ADAM17 siRNA was ineffective. Fractional inhibition of shedding was greatest for WT TREM2 using ADAM10 siRNA.

Source data are available online for this figure.

## Discussion

The role of neuroinflammation in neurodegenerative diseases is the focus of intense research effort, not least because of the genetic evidence pointing to the importance of the innate immune system in AD. Variants of TREM2 that are linked with Nasu–Hakola disease and/or AD appear to show loss of function phenotypes, perhaps rendering microglia less able to remove inflammatory material from the brain. While there is experimental support for TREM2 playing such a protective role during amyloid plaque deposition (Wang *et al*, 2015; Yuan *et al*, 2016), there are also conflicting accounts (Jay *et al*, 2015). The apparent experimental discordance is likely due to different age-related effects on plaque deposition in TREM2 knock-out mouse models (Jay *et al*, 2017). In addition to its role in phagocytosis, TREM2 also signals through its co-receptor DAP12, resulting in a cascade of phosphorylation mediated by ERK and Syk amongst other kinases (Xing *et al*, 2015). We know this signalling function is essential for health because mutations in DAP12 have also been linked to Nasu–Hakola disease (Paloneva *et al*, 2000). TREM2 signalling likely has anti-inflammatory consequences (Hamerman *et al*, 2005, 2006; Takahashi *et al*, 2005) and so any disease-linked downregulation would be expected to hamper resolution of an inflammatory reaction.

In this study, we have confirmed that the TREM2 sheddases in primary human macrophages and in primary murine microglia release TREM2 NTFs of a similar size into the culture supernatant. When deglycosylated, the predominant, larger species has a molecular mass of 17 kDa, while a smaller fragment of 15 kDa is also present. Concomitantly, the corresponding CTF accumulates in the cell lysate fraction. Using surface biotinylation, we observed that the turnover of TREM2 on the surface of macrophages is remarkably rapid, having a half-life of < 1 h, with most material being shed into the medium. Such rapid cycling predicts that a small change in either the rates of delivery to, or shedding from, the cell surface may result in significant changes in steady-state surface expression.

To quickly focus on the likely sheddase site, we used a tiled library of retro-inverso peptidomimetics that were structurally similar to the extracellular peri-membranous region of TREM2. We found mimetics that reduced shedding from HEK293 cells all included residues analogous to amino acids 158–160 (N'-SIS-C') from WT TREM2. The most effective inhibitor, C'-hvehsisrsll-N', was then synthesised as the corresponding reverse-sequence D-polypeptide, ostensibly as a negative control. Predictably, this reverse retro-inverso peptidomimetic will not conserve structure-specific side-chain interactions; however, general biophysical characteristics such as hydrophobicity and charge distribution will be conserved. The finding that both the forward and reverse peptidomimetics were equally potent at blocking shedding indicates that their activity likely depends on the biophysical combination of negative and positive charges in the peptide. Indeed, the glutamate–arginine pairing found in the mimetic resembles the ideal ADAM10 substrate proposed by Caescu and colleagues (Caescu *et al*, 2009).

We then used mass spectrometry to show that the sheddase cleavage site in human and mouse TREM2 occurs N-terminal to the tripeptide that we had identified, namely at the H157-S158 bond. This cleavage site is in the juxtamembrane region of the protein and is 17 amino acids from the predicted transmembrane domain. Such perimembrane cleavage is typical for α-secretase activity (Kleinberger

*et al*, 2014) and, as expected, the principal sheddase responsible for cleavage of wild-type TREM2 appears to be a metalloprotease, its activity being substantially blocked by GM6001, GI254023X and batimastat. By contrast, the sheddase was inhibited by neither the serine protease inhibitor PMSF nor the MMP2/9 inhibitor SB-3CT.

We have shown that the H157Y substitution, which is associated with increased risk of AD, results in accelerated cleavage and loss of TREM2, particularly the mature glycoforms, from the surface of HEK293 cells. Both the CTF and the shed NTF accumulate more rapidly in HEK293 cells expressing the variant as compared to wild-type TREM2. Mass spectrometric detection of shed peptides indicated that residue 157 is the most common C-terminal residue for both wild-type and H157Y TREM2. While GI254023X fully blocks the enhanced cleavage of H157Y TREM2, by contrast batimastat is less effective. This observation is consistent with the variant being susceptible to cleavage by one, or more, additional proteolytic enzymes. To investigate this further, we used siRNA to knock down either ADAM10 or ADAM17, or both. This experiment clearly showed that ADAM10 was the only enzyme of the two that was acting as a sheddase; however, the ADAM10 knock-down was more effective at preventing WT TREM2 shedding as compared to H157Y. Further work will be needed to determine whether the differential effects of batimastat and ADAM10 siRNA reflect structural alterations induced by the H157Y substitution, making it a preferred substrate for ADAM10. An alternative explanation is that the variant is a substrate for a completely independent protease.

While our results provide a molecular mechanism for how the H157Y variant might alter risk of AD, they do not presently shed light on whether the pathogenic effect of the accelerated cleavage could arise from more rapid removal of TREM2 from the cell surface and/or from the increased abundance of the released NTF. Regardless, our observations suggest that selective partial inhibition of cleavage of TREM2 at H157-Ser158 bond might provide a potential therapeutic strategy for carriers of the H157Y and possibly for individuals with wild-type TREM2. More generally, the increased CSF levels of soluble TREM2 in sporadic (Heslegrave *et al*, 2016; Piccio *et al*, 2016; Suárez-Calvet *et al*, 2016b) and familial AD (Suárez-Calvet *et al*, 2016a) argue that increased TREM2 NTF shedding is important in the common sporadic form of AD as well as in rare TREM2-dependent forms.

# Materials and Methods

### Macrophage cell culture

Primary human macrophages were differentiated from peripheral blood monocytes isolated from leucocyte cones (NHSBT, UK, samples obtained from healthy donors with informed consent according to the WMA Declaration of Helsinki and the Department of Health and Human Services Belmont Report) derived from male donors. Briefly, leucocyte cones were diluted to a volume of 40 ml in PBS and mononuclear cells isolated on polysucrose gradient. Cells were washed in PBS and plated in Corning flasks for 1 h in serum-free medium. Non-adherent contaminating cells were removed and remaining monocytes differentiated for 10 days with recombinant human GM-CSF (R&D Systems, 10 ng/ml) in RPMI with 10% v/v foetal bovine serum (FBS) and 1% w/v penicillin–streptomycin

(Invitrogen, Sigma, UK) and incubated at 37°C, 5% v/v $CO_2$ and 95% v/v $O_2$.

### Isolation and culture of neonatal microglia

Primary cultures of mixed murine glial cells were cultured from C57/BL6 P2 neonatal pups, purchased post-mortem from Charles River. Briefly, brains were extracted, rolled across sterile filter paper to remove vasculature and meninges and were mechanically dissociated. Cells were resuspended in 40 ml per 175 $cm^2$ flask (Corning, UK) and were maintained in Dulbecco's modified Eagle's medium (DMEM) with 10% v/v FBS and 1% w/v penicillin–streptomycin (Invitrogen, Sigma, UK) and incubated at 37°C (5% v/v $CO_2$/95% v/v $O_2$). A week later, the media was replaced containing recombinant murine GM-CSF (5 ng/ml, R&D systems, UK) and the cells were maintained for a further week. One week later, microglial cells were harvested by overnight shaking in an orbital shaker incubator (37°C, 180 rpm) and plated at 1–1.5 × 10^6/ml in 175 $cm^2$ flasks.

### Transfection of hTREM2 and hDAP12 into HEK293 cells and cleavage monitoring in HaloTag-expressing cells

Naïve HEK293 cells were transiently transfected with constructs expressing hDAP12 and hTREM2 (wild type or H157Y mutant fused or not at the N-terminus with a HaloTag). Twenty-four or 48 h after Lipofectamine 2000-mediated transfection, the culture medium was replaced and then conditioned for 5 h (HaloTag-TREM2) or 24 h (untagged TREM2) before collection. For stable expression, selection medium was added and cells grown as a stably expressing unsorted pool. Cell supernatants and lysates, prepared in Cell Lysis Buffer (Cell Signalling), were collected from cultures and stored at −80°C prior to analysis. Shed HaloTag-NTF and surface-exposed HaloTag-TREM2 were detected (Fig EV3) by including non-cell-permeant HaloTag ligand Alexa Fluor 660 (Promega) in the culture medium. Supernatants were subjected to SDS–PAGE, and proteins from the cell lysates were processed as described below.

### Immunocytochemistry

Cells were blocked in HBSS (Gibco), 20 mM HEPES (Gibco), 10% v/v donkey serum (Abcam) (20 μg/ml Fc fragment was added for human macrophages, Rockland). Cells received primary antibodies (AF1828, goat anti-hTREM2 N-terminal domain, from R + D Systems, 2 μg/ml), spiked into the above block buffer for 1 h on ice. Control experiments used naïve goat serum as the primary antibody. Cells were rinsed twice with ice-cold HBSS and received donkey anti-goat Alexa Fluor 546 or 647 (Life Technologies) at 1:1,000 in blocking buffer for 30 min on ice. Cells were rinsed with ice-cold PBS and fixed in 4% v/v paraformaldehyde for 10 min at room temperature. Hoechst (Life Technologies) was added to wells in PBS and cells visualised on an Olympus IX81 fluorescence microscope.

### Protein deglycosylation

TREM2, from conditioned media or immunoprecipitated material, was deglycosylated using Protein Deglycosylation Mix (New England

Biolabs) as per manufacturer's instructions. Briefly, the sample was mixed with glycoprotein denaturing buffer and heated to 95°C for 5 min. GlycoBuffer, NP40 and deglycosylation enzyme cocktail were added, and the reaction was incubated for 18 h at 37°C.

### siRNA

Human embryonic kidney cells stably expressing WT or H157Y variant hTREM2 and hDAP12 were plated in 24-well plates (Costar) at 50,000 cells per well; 24 h later, cells received control siRNA (50 nM) or siRNA to ADAM10 or ADAM17 (25 nM each) or both ADAM10 + ADAM17 siRNA (25 nM of each). The siRNA was in smart pool format from Dharmacon, made up as stated by the manufacturer and pre-mixed with DharmaFECT transfection reagent prior to addition to cell cultures. Details of the siRNA used in this study are as follows: control siRNA: ON-TARGET plus Non-targeting Pool (D-001810-10-05); ADAM10 siRNA: ON-TARGET plus Human ADAM10 (102) siRNA SMART pool (L-004503-00-0005); ADAM17 siRNA: ON-TARGET plus Human ADAM17 (6868) siRNA SMART pool (L-003453-00-0005). The transfection reagent containing siRNA was left on the cells for 72 h at which point conditioned media and cell lysates were collected and frozen at −80°C prior to TREM2 quantification by MSD assay.

### Gel-based protein visualisation: Western blotting

Cell supernatants or lysates (prepared in Cell Lysis Buffer, Cell Signaling) were mixed with NuPAGE LDS Sample Buffer and sample reducing agent (Life Technologies) and run on 4–12% w/v Bis–Tris gels (Life Technologies). Gels were transferred to PVDF membrane, which was probed with AF1828, goat anti-hTREM2 N-terminal domain or MAB1729, rat anti-mTREM2-N-terminal domain (both from R&D Systems, 2 μg/ml in Odyssey block or 5% w/v non-fat milk).

Blots in the siRNA experiments were probed with polyclonal rabbit anti-ADAM10 at 1:500 (14194S, Cell Signaling), rabbit anti-ADAM17 at 1:500 (AB19027, Millipore) and mouse anti-β-actin at 1:1,000 (ab3280, Abcam). Immunocomplexes were detected using the following secondary antibodies all at 1:10,000: rabbit anti-mouse (Thermo Scientific), rabbit, mouse or goat IgG HRP-conjugated antibody (R&D Systems), IRDye® 680RD donkey anti-goat IgG or IRDye® 800CW goat anti-rat IgG. Molecular weight ladders used were Chameleon Duo Pre-stained Protein Ladder (LI-COR) or MagicMark™ XP (ThermoFisher). Blots were either visualised on LI-COR Odyssey imaging system or using photographic film.

### Treatment with peptides and small molecule antagonists

Peptidomimetic retro-inverso peptides: C'-hvehsis-N', C'-sisrsll-N', C'-hvehsisrsll-N' & C'-fedahvehsis-N', C'-sllegei-N', C'-geipfpp-N' & C'-fpptsil-N', were synthesised by (Cambridge Research Biochemicals) and diluted in cell culture medium to concentrations of 5 mM. The broad-spectrum metalloprotease inhibitors GM6001 (Tocris) and batimastat (Sigma), the ADAM10 inhibitor GI254023X (Tocris), the serine protease inhibitor PMSF (Sigma) and the MMP2/9 inhibitor SB-3CT (Sigma) were diluted in cell culture medium at concentrations between 10 nM and 50 μM.

### The paper explained

#### Problem

Neuroinflammation has been implicated in the pathogenesis of Alzheimer's disease by genomic studies. Specifically, the role of the brain's phagocytic cells, the microglia, has been highlighted. TREM2 is a receptor protein that allows microglia to sense and potentially engulf the amyloid plaques and cellular debris that characterise Alzheimer's disease. The delivery of TREM2 to the cell surface is impeded by variants that cause neurodegenerative disease (Nasu–Hakola disease). It is less clear whether variants linked to Alzheimer's disease affect the levels of TREM2 at the cell surface. The balance of delivery and proteolytic shedding of TREM2 from the cell surface may have a role in the neuroinflammatory pathogenesis of Alzheimer's disease.

#### Results

TREM2 expression on, and shedding from, human and mouse primary myeloid cells and from human embryonic kidney cells was measured. The sheddase activity was characterised using a range of protease inhibitors and it was concluded that for wild-type TREM2 ADAM10, a promiscuous metalloproteinase, was responsible, consistent with the literature. The site of sheddase cleavage was determined (between positions 157 and 158), initially using peptidomimetic inhibitors and then by mass spectrometry. The disease-linked H157Y variant of TREM2 was found to be shed more rapidly from cells; accelerated loss might be due to the recruitment of a novel proteolytic activity.

#### Impact

The sheddase site identified in TREM2 lies between the folded extracellular domain of TREM2 and the outer surface of the cell membrane. From our study, the H157Y mutation at the sheddase site likely increases risk of Alzheimer's disease by accelerating proteolytic loss of TREM2 from the cell surface. The sheddase site is available to potential therapeutic reagents, such as antibodies, that might protect TREM2, favouring its retention on the microglial surface. Understanding the balance of delivery and loss of TREM2 at the cell surface might allow us to develop novel therapies to regulate neuroinflammation in Alzheimer's disease.

### Immunoprecipitation of TREM2 for mass spectrometry

The cell culture medium of macrophages, microglia or HEK293 (overexpressing TREM2) was conditioned for up to 7 days, spiked with complete protease inhibitor cocktail (Roche) and 20 mM HEPES (Gibco) before concentrating 10-fold on molecular weight cut-off Amicon Ultra-15 3 kDa centrifugation devices. Concentrated medium from human cells (HEK293, macrophages) or mouse microglia were incubated with goat anti-human TREM2, or rat anti-mouse TREM2, respectively (both 10 μg/ml from R&D Systems) and immunoprecipitated using Dynabeads® Protein G (ThermoFisher). TREM2 was eluted in 10 mM glycine, pH 1.5, and neutralised with 1.5 M Tris, pH 8.0.

### TREM2 meso scale discovery immunoassay

A sandwich immunoassay quantified the concentrations of shed TREM2 in cell culture supernatants from macrophages and HEK293 cells over-expresssing human TREM2. Rabbit anti-TREM2 monoclonal capture antibody from Sino Biological's ELISA was coated at 2 μg/ml on MSD L15XA plates. Plates were washed in 0.05% v/v Tween 20 in PBS and blocked with 3% w/v BSA. Samples and a 7-point standard curve with recombinant human

TREM2 (Sino Biological) diluted in cell culture medium were generated and added to plates for 2 h at room temperature. Plates were washed and each well received detection antibody (biotin-conjugated goat anti-TREM2 from R&D, BAF1828, at 1 μg/ml in MSD Diluent 100). Following a 1-h incubation at room temperature, plates were washed and wells treated with 1:500 streptavidin–SULFO-TAG. The SULFO-TAG emits light when immobilised on the electrode in each well and this signal is detected by a charge-coupled device camera (MSD, www.mesoscale.com). Plates were incubated at RT for 1 h, washed and read on the MSD SECTOR Imager 6000. Validation of the MSD assay, showing TREM2 specificity for both lysates and supernatants, is shown in Fig EV4.

Briefly, plates were coated with anti-TREM2 antibody as above. Samples for the standard curve were generated using biotinylated recombinant TREM2. Experimental samples containing biotin-TREM2 were derived from the surface-biotinylation experiments. Both types of samples were detected by the addition of streptavidin-SULFO-TAG.

## Surface biotinylation and cell fractionation

Primary cultures of human macrophages were surface-biotinylated using EZ-Link Sulfo-NHS-LC-Biotin (ThermoFisher). Following surface biotinylation, cells were incubated at 37°C for 0–5 h. Supernatants and cells were collected over a time course. At each time point, cells were fractionated into membrane, organelle, nuclear and cytosolic components using Minute™ Plasma Membrane Protein Isolation Kit (Invent Biotechnologies). Levels of biotin-TREM2 in each of the samples were measured by biotin-TREM2 MSD assay. Organelle fractions were contaminated with plasma membrane proteins, both TREM2 and marker proteins, and so no attempt was made to tabulate these two signals separately.

## Silver gel purification and mass spectroscopy of protein samples

Immunoprecipitated and deglycosylated TREM2 from cell cultures was mixed with NuPAGE LDS Sample Buffer and sample reducing agent (Life Technologies) and run on 4–12% w/v Bis–Tris gels (Life Technologies). Following electrophoresis, gels were stained using Pierce's silver stain (mass spectrometry compatible) according to manufacturer's instructions (ThermoFisher).

The gel bands were transferred into a 96-well PCR plate cut into 1-mm$^2$ pieces, destained, reduced (DTT) and alkylated (iodoacetamide) and subjected to enzymatic digestion with trypsin overnight at 37°C. After digestion, the supernatant was loaded onto an autosampler for automated LC-MS/MS analysis.

Mass spectrometry was performed primarily at the Cambridge Centre for Proteomics, University of Cambridge, Cambridge, UK, and also at the Clinical Proteomics Mass Spectrometry SciLifeLab, Karolinska Institutet, Sweden (specifically Fig 4B and C). LC-MS/MS experiments were performed using a Dionex Ultimate 3000 RSLC nanoUPLC (Thermo Fisher Scientific Inc, Waltham, MA, USA) system and a Q Exactive Orbitrap mass spectrometer (Thermo Fisher Scientific Inc). Peptides were separated by reverse-phase chromatography at a flow rate of 300 nl/min and a Thermo Scientific reverse-phase nano Easy-spray column (Thermo Scientific PepMap C18, 2 μm particle size, 100 Å pore size, 75 μm internal diameter × 50 cm length). Peptides were loaded onto a pre-column (Thermo Scientific PepMap 100 C18, 5 μm particle size, 100 Å pore size, 300 μm internal diameter × 5 mm length) from the Ultimate 3000 autosampler with 0.1% v/v formic acid for 3 min at a flow rate of 10 μl/min. The column valve was then switched to elute peptides from the pre-column onto the analytical column. Solvent A was water + 0.1% v/v formic acid, and solvent B was 80% v/v acetonitrile, 20% v/v water + 0.1% v/v formic acid. The linear gradient applied was 2–40% B over 30 min.

The LC eluant was sprayed into the mass spectrometer by means of an Easy-Spray source (Thermo Fisher Scientific Inc.). All m/z values of eluting ions were measured in an Orbitrap mass analyser, set at a resolution of 70,000 and were scanned between m/z 380 and 1,500. Data-dependent scans (Top 20) were employed to automatically isolate and generate fragment ions by higher energy collisional dissociation (HCD, NCE:25%) in the HCD collision cell, and measurement of the resulting fragment ions was performed in the Orbitrap analyser, set at a resolution of 17,500. Singly charged ions and ions with unassigned charge states were excluded from being selected for MS/MS, and a dynamic exclusion window of 20 s was employed.

Post-run, the data were processed using Protein Discoverer (version 2.1., ThermoFisher). Briefly, all MS/MS data were converted to mgf files and the files were then submitted to the Mascot search algorithm (Matrix Science, London, UK) and searched against a customised UniProt human database (176,496 sequences, 61,514,126 residues) or a customised UniProt mouse database (88,085 sequences, 37,920,032 residues). Each database contained a common contaminant sequences (115 sequences, 38,274 residues; http://www.thegpm.org/crap/) and modified versions of the extracellular domain of TREM2 sequence in both mouse and human versions. Briefly, a series of sequences were generated in which amino acids were removed sequentially from the C-terminal end of the sequences past the predicted sheddase site (H157 residue) and up to the S147 residue. A total of 29 (human) or 25 (mouse) new sequences were generated. Similar sequences were also generated for the H157Y mutant version of TREM2. Variable modifications of oxidation (M), deamidation (NQ) and carbamidomethyl were applied. The peptide and fragment mass tolerances were set to 5 ppm and 0.1 Da, respectively. A significance threshold value of $P < 0.05$ and a peptide cut-off score of 20 were also applied.

## Experimental design and statistical methods

There was insufficient prior knowledge of the data to be able to undertake formal power calculations. A minimum of three biologically independent experiments were undertaken. The variance of the data was assessed as it was acquired and the number of required repeats was adjusted accordingly. The range of experimental repeats was 3–15. The data were assumed to be normally distributed and that variance was similar between comparators. These assumptions were not explicitly tested.

For cell biology experiments, the investigators were not blinded. Where antibodies were employed, we used species and, where appropriate, isotype-matched, antisera as controls for specific antibody binding. Parental HEK293 cells were used as controls for hTREM2-expressing HEK293 cells. Where comparisons are made across a Western blot, all samples were run on SDS–PAGE, blotted,

probed and developed simultaneously. Peptidomimetics that were adjacent to the position 157–158 cleavage site were considered controls for the remainder. The reverse-sequence inhibitory peptidomimetic controlled for the effect of peptides with similar biophysical characteristics. The mass spectroscopy investigator (MD) was blinded. We did not randomise experiments.

**Expanded View** for this article is available online.

## Acknowledgements

We thank Ekaterina Rogaeva for reading the manuscript and helpful discussions. We acknowledge the services of the Clinical Proteomics Mass Spectrometry SciLifeLab, Karolinska Institutet, Sweden. We acknowledge helpful discussions with Prof. Christian Haass, Munich. The initial conversation took place at the kick-off meeting of PHAGO Innovative Medicines Initiative 2 Joint Undertaking under grant agreement No 115976, which receives support from the European Union's Horizon 2020 research and innovation programme and EFPIA. Funding from the Wellcome Trust and the Canadian Institutes of Health Research contributed to the support of this study.

## Conflict of interest

The authors declare that they have no conflict of interest. PT, GF, SS, EHF, RBD, LSCN, SE, JR, AB, IC and DCC and are employees of AstraZeneca or its subsidiaries.

## Author contributions

PT, JS, MJD, GF, YZ, SS, EHF, BGP-N and LSCN performed the experiments and analysed the data. RBD, SQ, SE, JR, CJ, AB, PHSG-H, IC and DCC conceived and supervised the experiments. PT, JS, PHSG-H and DCC wrote the manuscript.

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
