## [Review Process File · EMBO Molecular Medicine]

TREM2 shedding by cleavage at the H157-S158 bond is accelerated for the Alzheimer's disease-associated H157Y variant

Peter Thornton, Jean Sevalle, Mike Deery, Mr. Graham Fraser, Ye Zhou, Sara Ståhl, Elske Franssen, Roger Dodd, Seema Qamar, Beatriz Perez-Nievas, Louise Nicol, Susanna Eketjäll, Revell Jefferson, Clare Jones, Andrew Billinton, Peter St George-Hyslop, Iain Chessell, Damien C Crowther

Corresponding author: Damian Crowther, AstraZeneca Innovative Medicines and Early Development

Review timeline:

Submission date:	07 February 2017
Editorial Decision:	15 March 2017
Revision received:	27 June 2017
Editorial Decision:	18 July 2017
Revision received:	31 July 2017
Accepted:	04 August 2017

Transaction Report:

Editor: Céline Carret

1st Editorial Decision

15 March 2017

Thank you for the submission of your manuscript to EMBO Molecular Medicine. We have now heard back from the three referees whom we asked to evaluate your manuscript. Although the referees find the study to be of potential interest, they also raise a number of concerns that must be addressed in the next final version of your manuscript.

As you will see from the comments below, that referee 2 is somehow concerned by the quality of western blots in figures 1 and 2, a titration of inhibitor that is missing along with the use of shRNAs to validate the specificity, and the concept of using the peptidomimetic that should be better explained. Overall referees 1 and 2 request additional explanations, and clarifications that we agree are needed.

Given the balance of these evaluations, we feel that we can consider a revision of your manuscript if you can address the issues that have been raised within the space and time constraints outlined below. Please note that it is EMBO Molecular Medicine policy to allow only a single round of revision and that, as acceptance or rejection of the manuscript will depend on another round of review, your responses should be as complete as possible.

EMBO Molecular Medicine has a "scooping protection" policy, whereby similar findings that are

published by others during review or revision are not a criterion for rejection. Should you decide to submit a revised version, I do ask that you get in touch after three months if you have not completed it, to update us on the status.

I look forward to receiving your revised manuscript.

***** Reviewer's comments *****

Referee #1 (Remarks):

In this manuscript by Thornton et al., the authors characterize the proteolytic cleavage events responsible for shedding of TREM2 in primary cultures of human macrophages, murine microglia and TREM2-expressing human embryonic kidney (HEK293) cells. They also demonstrated the NTF domains released into the media can be blocked using broad range metalloprotease inhibitors. They also showed that this cleavage occurs at specific residues in the Histidine 157 and serine 158 bond. This paper addresses the hypothesis in a clear and straightforward approach. However, there are several points the authors should address to improve their work.

Minor comments:

1. The authors used neonatal murine microglia. Since previous studies have validated that primary and adult microglia have different genetic and protein profiles, the expression of TREM2 and cleavage might be different in early development and adulthood in mice. It would be more beneficial of the authors to use adult microglia.
2. Human data: Are the donors male or female? Is there a selection criterion? This is an important factor to take into account considering gender difference have been shown.
3. In Figure 1 authors should consider inserting a schematic diagram of TREM2 showing the N-glycosylation sites and the H157-S158 bond, this would allow for the readers to clearly visualize where the authors propose TREM2 NTF is cleaved.
4. In Figure 1, the authors demonstrated that TREM2 is mostly released into the extracellular space and not localized in the cytoplasm or nucleus. The title of the figure should be reconsidered, as the term trafficking might be inappropriate?
5. Figure 4 should be rearranged- put the quantification of WB beside the blot for ease of following the data. Legend should include all panels of the figure in order.
6. Did authors look at higher concentrations of inhibitors? What happens when one treated with 100?
7. Why did the authors treat with only GI254023X and Batimastat (Sigma) and not GM6001 (Tocris)?
8. Figure 4E,F: Label the blot and graphs with correct concentrations e.g. 0nM, 0.1 nM 10 nM etc. it is not clear what concentrations were used to treat the samples. mM, uM or nM? What is the biological toxicity of these inhibitors, did author carry out toxicity/kinetics studies on these cells?
9. Figure 4: Did authors try specific MMP inhibitors? e.g. MMP9?
10. Figure 5: Minor editorial adjustment to the figure to fit in the box. Also Label clearly on the B) which peptide mimetic is the control for example over red and blue bars label mimetic vs. control.
11. Figure 5B: What control was used?

Referee #2 (Remarks):

This manuscript addresses the cell biological properties of a mutant form of the Triggering Receptor Expressed on Myeloid Cells 2 (TREM2) that has been implicated in the pathogenesis of Alzheimer's disease (AD). The authors show that the H157Y mutant of TREM2 is shed more efficiently from the cell surface than the wild type protein. Based on this finding, they suggest that the decreased surface levels of the mutant TREM2 could affect the function of macrophages and microglia to act as

scavengers and remove Abeta peptides, thereby contributing to the development of AD. Consequently, blocking TREM2 shedding might be considered a novel treatment for AD. Overall, studies on the role of TREM2 in AD have emerged as an exciting area of research in AD. However, in this reviewer's opinion, the quality of the data in several figures makes their interpretation difficult, and there are conceptual problems with the peptide inhibition experiments. Significant changes, including additional experiments would be required to improve the quality and potential impact of this manuscript,

As to specifics:

- 1) The quality of the Western blots in figure 1B and 2A should be improved. Essentially, the quality of the Western blot in figure 4 is much better, and all experiments should be performed in this manner. The weak "smear" in the macrophage lane in figure 2A is an example of results that are not particularly convincing as shown. This does not necessarily cast doubt on the overall interpretation, but for a high-quality journal such as Embo MM, this reviewer would expect to see uniformly high quality data.
- 2) The authors should show the results they used for the quantification shown in Figure 1 C.
- 3) The MSD assay should be briefly described in the results, and the name spelled out.
- 4) Regarding figure 4, it is not clear why the authors used such a large jump in inhibitor concentration in E (from 0.1 to 10 μ M)? This is not particularly informative, and there should be more intermediate data points (e.g. systematic 2 or 3-fold dilutions of the inhibitor concentration over this range). Moreover, siRNA knockdown of the most likely candidates, i.e. ADAM10 and ADAM17 should be used to test the unspoken notion that ADAM10 could be involved in shedding TREM2 (as implied by the use of the somewhat ADAM10-selective GI254023X inhibitor, albeit at non-selective concentrations). Please note that ADAM10 is efficiently blocked by 1 μ M of the GI inhibitor. Based on these data, the TREM2 sheddase should be a different protease, such as ADAM17, although previous studies have implicated ADAM10 as the key sheddase. These findings are not discussed or mentioned, which can be considered a major omission for a paper that is focused on TREM2 shedding. The authors should also include treatment of cells with 25 ng/ml of PMA for 45 minutes (which activates ADAM17) and treat macrophages with different concentrations of LPS for 1 and 2 hours to determine how this affects TREM2 shedding. Then they should discuss their results in the context of what is known about the metalloproteases responsible for TREM2 shedding. Finally, the minor difference in BB94 inhibition shown in Figure 4F and supplementary figure 2 G is not sufficient to propose other protease activities without showing a better inhibitor titration, and siRNA experiments. The mutation might simply make the H157Y mutant a better substrate, and thus perhaps a bit more difficult for a competitive inhibitor to block.
- 5) The data with high concentrations of a peptidomimetic protease inhibitor in figure 5 are not particularly convincing or informative. It is not clear whether or not the peptides shown here are selective inhibitors, based on the limited data shown here, where high inhibitor concentrations elicit a small effect, and two out of the three control peptides seem to activate shedding. If the >fpptsil< peptide is used as a reference point, do the peptides mimicking the cleavage site show any significant effect? In addition, the stated premise of these experiments is flawed, since it is not clear how very high concentrations of a peptidomimetic inhibitor would have advantages over a metalloprotease inhibitor in terms of substrate selectivity or side effects? If, for example, ADAM10 or ADAM17 is the major TREM2 sheddase, and the authors would like to avoid using a hydroxamate to block a pleiotropic enzyme, how would adding a peptidomimetic solve this problem? If it works as proposed, a competitive inhibitor should bind to the catalytic site of the responsible enzyme, which would block processing of other targets of the enzyme as well.
- 6) The "Gel based protein visualization" section in the methods is mostly duplicated and should be consolidated into one section.

Referee #3 (Comments on Novelty/Model System):

The authors provide a first report of the TREM2 cleavage site, which also corresponds to a known

pathogenic AD-related mutation, making these findings highly medically relevant. The author's use of primary microglia and human monocytic cells, along with a cell line, is appreciated and ideally models the system in question, to the extent experimentally possible. Multiple experimental methods are used to confirm the findings presented, and all experiments are performed with sufficient replication and are properly controlled. Statistical analyses are sufficient.

Referee #3 (Remarks):

The authors have presented a timely, highly relevant and interesting finding. All experiments were performed with sufficient rigor to support the claims presented and detailed in a manner sufficient to allow reproduction by others. The use of primary microglia and human monocytic cells is greatly appreciated and represents the best possible modeling of the system in question.

The use of peptidomimetic protease inhibitors proved an innovative strategy that demonstrates potential for future therapeutic avenues, adding interest to the overall body of work presented. The authors did not attempt to test functional consequences of the mutant variant in their study, however, it is the opinion of this reviewer that the presented findings are sufficient to warrant publication without functional data and that the assessment of functional consequences warrants a significant amount of work that is beyond the scope of this study.

I recommend publication without modifications.

1st Revision - authors' response

27 June 2017

Reviewer 1:

"1. The authors used neonatal murine microglia. Since previous studies have validated that primary and adult microglia have different genetic and protein profiles, the expression of TREM2 and cleavage might be different in early development and adulthood in mice. It would be more beneficial of the authors to use adult microglia."

This data would be nice to have; while we have results from adult cells (primary human macrophages) and microglial cells (primary neonatal mouse microglia), we do not have cells that are both murine and adult-derived. We have examined the feasibility of undertaking mass spectroscopy experiments to characterise the shedding in primary adult mouse microglia but, considering the small yield of cells from each mouse brain, we would need to sacrifice many dozen animals. This is something that we are not able to do. We propose that the adult macrophage is a reasonable model of TREM2 processing in adult microglia – not least because of the similar results seen across the two primary cell types and the HEK293 cells.

"2. Human data: Are the donors male or female? Is there a selection criterion? This is an important factor to take into account considering gender difference have been shown."

The primary human macrophages were derived from male donors and the HEK293 cells were genetically female. The murine primary microglia were prepared from pooled extracts of brains from both male and female animals. The sheddase site is consistently at His157-Ser158 across all three cell preparations and this argues against there being marked sex differences. To make this explicit we have included the following text in the "Macrophage cell culture" section of the Methods:

"...monocytes isolated from leukocyte cones ... derived from male donors."

"3. In Figure 1 authors should consider inserting a schematic diagram of TREM2 showing the N-glycosylation sites and the H157-S158 bond, this would allow for the readers to clearly visualize where the authors propose TREM2 NTF is cleaved."

This is a great suggestion and we have added the following schematic to the manuscript as fig. 4E:

“4. In Figure 1, the authors demonstrated that TREM2 is mostly released into the extracellular space and not localized in the cytoplasm or nucleus. The title of the figure should be reconsidered, as the term trafficking might be inappropriate?”

We agree with this comment and have changed the title to: “Figure 1: TREM2 expression, glycosylation and proteolysis.”

“5. Figure 4 should be rearranged- put the quantification of WB beside the blot for ease of following the data. Legend should include all panels of the figure in order.”

We agree with the reviewer and have rearranged the panels as suggested. There is inevitably some cross-referencing from panel A to both panel B (quantitation of full length TREM2 isoforms) and also to panel D (quantitation of proteolytic fragments of TREM2), however the clarity has been improved. We have also included a schematic showing how GI254023X and batimastat are expected to inhibit proteases; this will help the reader interpret panel F and anticipates the point below.

“6. Did authors look at higher concentrations of inhibitors? What happens when one treated with 100?”

We did not add inhibitors at higher concentrations in the experiments shown in Fig. 4 because 10 mM is already several fold higher than the IC_{50} for all the inhibitors against all the likely proteases (except for GI254023X which does not inhibit MMPs). In subsequent experiments, we have titrated batimastat up to 30 mM in HEK293 culture to differentiate the behaviour of WT and H157Y TREM2. This data is presented in Fig. EV2:

At least for the WT 10 mM essentially all the inhibition is seen by 3-10 mM batimastat.

“7. Why did the authors treat with only GI254023X and Batimastat (Sigma) and not GM6001 (Tocris)?”

We used the combination of GI254023X and Batimastat because of their complementary activity against ADAM10 and ADAM17. GI254023X has a higher potency against ADAM10 and conversely batimastat has greater potency for ADAM17. GM6001 would have inhibited both ADAM10 and 17 equally, something that we also achieved with 10 mM of our inhibitors.

“8. Figure 4E,F: Label the blot and graphs with correct concentrations e.g. 0nM, 0.1 nM 10 nM etc. it is not clear what concentrations were used to treat the samples. mM, uM or nM? What is the biological toxicity of these inhibitors, did author carry out toxicity/kinetics studies on these cells?”

We thank the reviewer for the feedback about clarity. We have included comments about concentrations of inhibitors and the mass of gel markers in the legend and annotated the figures as suggested. We did undertake toxicity assays against the HEK293 cells used in the manuscript and showed that, for concentrations as high as 10 mM, there was no change in the MTT signal for any of the protease inhibitors. We did not undertake kinetic studies with the protease inhibitors.

“9. Figure 4: Did authors try specific MMP inhibitors? e.g. MMP9?”

Yes, in response to this comment, we repeated the protease inhibitor treatment using HEK293 cells (new Fig. 2C). We show that in these cells, GM6001 and GI254023 prevent shedding but PMSF and the MMP2/9 inhibitor, SB-3CT, do not at concentrations up to 30 mM. This is reported in the results:

“Similar effects on shedding were seen in HEK293 cells treated with these protease inhibitors; additionally, the MMP2/9 inhibitor SB-3CT did not inhibit shedding (fig. 2C).”

“10. Figure 5: Minor editorial adjustment to the figure to fit in the box. Also Label clearly on the B) which peptide mimetic is the control for example over red and blue bars label mimetic vs. control.” and

“11. Figure 5B: What control was used?”

We have ensured that the figure fits within the box and that we have increased the size of the labels to help with clarity.

Regarding controls for the peptide mimetic experiments: in the original draft of the manuscript we used peptide mimetics that *did not include the cleavage site* as controls for those that did, hence the pairwise comparison in panel C. In response to this comment, we have repeated the experiment, taking the most effective inhibitor (C'-hvehsirsll-N') and comparing it with its reverse sequence peptide mimetic, as a control. As expected, at 5 mM the peptide mimetic caused a 40% reduction in shedding, however the reverse sequence peptide also exerted a similar inhibitory effect. This indicates that, rather than being strictly sequence specific, the protease likely has a preference for substrates with particular biophysical characteristics. To our eyes the C'-hvehsirsll-N' peptide mimetic presents a negatively charged glutamate close to a positively charged arginine in the context of a generally uncharged/hydrophobic context. These characteristics resemble the optimal ADAM10 substrate as proposed by Caescu and colleagues (Caescu et al., 2009). This lends some support to the proposal that the major TREM2 sheddase is ADAM10. This interpretation of the data is now included in the the manuscript:

In the results:

“To understand whether the protease inhibition was strictly sequence specific, we synthesised a D-polypeptide with the reverse sequence of the most effective inhibitor (C'-hvehsirsll-N'). This reverse retro-inverso peptide was equally effective at preventing TREM2 shedding (fig. 3D), indicating that access to the protease is determined less by the specific amino acid sequence as by general biophysical characteristics such as charge.”

In the discussion:

“To quickly focus on the likely sheddase site we used a tiled library of retro-inverso peptidomimetics that were structurally similar to the extracellular peri-membranous region of TREM2. We found mimetics that reduced shedding from HEK293 cells all included residues analogous to amino acids 158-160 (N'-SIS-C') from WT TREM2. The most effective inhibitor, C'-hvehsirsll-N', was then synthesised as the corresponding reverse sequence D-polypeptide, ostensibly as a negative control. Predictably this reverse retro-inverso peptidomimetic will not conserve structure-specific side-chain interactions, however general biophysical characteristics such as hydrophobicity and charge distribution will be conserved. The finding that both the forward and reverse peptidomimetics were equally potent at blocking shedding indicates that their activity likely depends on the biophysical combination of negative and positive charges in the peptide. Indeed the glutamate-arginine pairing found in the

mimetic resembles the ideal ADAM10 substrate proposed by Caescu and colleagues (Caescu *et al*, 2009).”

Reviewer 2

“(1) The quality of the Western blots in figure 1B and 2A should be improved. Essentially, the quality of the Western blot in figure 4 is much better, and all experiments should be performed in this manner. The weak “smear” in the macrophage lane in figure 2A is an example of results that are not particularly convincing as shown. This does not necessarily cast doubt on the overall interpretation, but for a high-quality journal such as *Embo MM*, this reviewer would expect to see uniformly high quality data.”

We are grateful to the reviewer for this comment. We re-optimised our methods, finding that the blocking of the blots could be improved. We now have a fresh set of western blots that have a much lower background. We trust that these are now of sufficient quality for publication in *Embo MM*.

“(2) The authors should show the results they used for the quantification shown in Figure 1 C”

We have included the raw data in fig. EV1.

“(3) The MSD assay should be briefly described in the results, and the name spelled out.”

The MSD acronym is now expanded when first used (in the results section) and the methods section includes the following brief description and URL:

“Following a 1 h incubation at room temperature, plates were washed and wells treated with streptavidin-Sulfo-Tag. The Sulfo-Tag emits light when immobilised on the electrode in each well and this signal is detected by a charge coupled device camera (MSD, www.mesoscale.com).”

“(4) Regarding figure 4, it is not clear why the authors used such a large jump in inhibitor concentration in E (from 0.1 to 10 μ M)? This is not particularly informative, and there should be more intermediate data points (e.g. systematic 2 or 3-fold dilutions of the inhibitor concentration over this range).”

We agree that the 0.1 mM is likely to incompletely inhibit the target proteases and so we only quantify the 10 mM conditions where we are confident that this will cause full inhibition. We have undertaken full titrations; we present the data for batimastat (fig. EV2, shown above) to support the message that batimastat is less potent at preventing shedding when the cells express the H157Y variant as compared to WT TREM2.

... “Moreover, siRNA knockdown of the most likely candidates, i.e. ADAM10 and ADAM17 should be used to test the unspoken notion that ADAM10 could be involved in shedding TREM2 (as implied by the use of the somewhat ADAM10-selective GI254023X inhibitor, albeit at non-selective concentrations).”

We are grateful for this comment and in response we have undertaken siRNA knock-down experiments in the HEK293 cells, targeting ADAM10 and ADAM17 alone and in combination. This data now constitutes the new figure 6. There is a substantial additional section of the discussion that includes reference to this new data:

“While GI254023X fully blocks the enhanced cleavage of H157Y TREM2, by contrast batimastat is less effective. This observation is consistent with the variant being susceptible to cleavage by one, or more, additional proteolytic enzymes. To investigate this further we used siRNA to knock down either ADAM10 or ADAM17, or both. This experiment clearly showed that ADAM10 was the only enzyme of the two that was acting as a sheddase; however the ADAM10 knock down was more effective at preventing WT TREM2 shedding as compared to H157Y. Further work will be needed to determine whether the differential effects of batimastat and ADAM10 siRNA reflect structural alterations induced by the H157Y substitution, making it a preferred substrate for ADAM10. An alternative explanation is that the variant is a substrate for a completely independent protease.”

... “Please note that ADAM10 is efficiently blocked by 1 μ M of the GI inhibitor.”

We accept this and only the 10 mM conditions are quantified.

... “Based on these data, the TREM2 sheddase should be a different protease, such as ADAM17, although previous studies have implicated ADAM10 as the key sheddase. These findings are not discussed or mentioned, which can be considered a major omission for a paper that is focused on TREM2 shedding.”

This deficit in the manuscript has been remedied as described above.

... "The authors should also include treatment of cells with 25 ng/ml of PMA for 45 minutes (which activates ADAM17) and treat macrophages with different concentrations of LPS for 1 and 2 hours to determine how this affects TREM2 shedding."

The work presented here looks only at basal conditions for the macrophages. The consequences of activation for TREM2 biology are of course important but we feel that they are beyond the scope of this manuscript; instead we maintain our focus on basal proteolytic processing of TREM2 and any differences between WT and the H157Y variant.

... "Then they should discuss their results in the context of what is known about the metalloproteases responsible for TREM2 shedding. Finally, the minor difference in BB94 inhibition shown in Figure 4F and supplementary figure 2 G is not sufficient to propose other protease activities without showing a better inhibitor titration, and siRNA experiments. The mutation might simply make the H157Y mutant a better substrate, and thus perhaps a bit more difficult for a competitive inhibitor to block."

We now have all the components requested. In addition to fig. 5 we now have data showing the consequence of siRNA targeting ADAM10 and ADAM17. This indicates that ADAM10 knock-down provides for more complete shedding blockade in cells expressing WT TREM2 as compared to the variant. In the discussion we state:

"...the ADAM10 knock down was more effective at preventing WT TREM2 shedding as compared to H157Y."

Furthermore the full titration of the batimastat effect (fig. EV2) confirms that the variant TREM2 exhibits higher levels of batimastat-resistant shedding.

"5) The data with high concentrations of a peptidomimetic protease inhibitor in figure 5 are not particularly convincing or informative. It is not clear whether or not the peptides shown here are selective inhibitors, based on the limited data shown here, where high inhibitor concentrations elicit a small effect, and two out of the three control peptides seem to activate shedding. If the >fpptsil< peptide is used as a reference point, do the peptides mimicking the cleavage site show any significant effect? In addition, the stated premise of these experiments is flawed, since it is not clear how very high concentrations of a peptidomimetic inhibitor would have advantages over a metalloprotease inhibitor in terms of substrate selectivity or side effects? If, for example, ADAM10 or ADAM17 is the major TREM2 sheddase, and the authors would like to avoid using a hydroxamate to block a pleiotropic enzyme, how would adding a peptidomimetic solve this problem? If it works as proposed, a competitive inhibitor should bind to the catalytic site of the responsible enzyme, which would block processing of other targets of the enzyme as well."

We have taken these points into consideration and now step back from some of the earlier claims, in particular, we no longer claim that peptidomimetic approaches have advantages over other inhibitors of ADAM10. Instead we are now presenting the mimetic data earlier in the manuscript, to reflect its real utility for quickly locating the site of sheddase cleavage. It is a matter of fact that the three amino acid "region of interest" generated by this approach was adjacent to the cleavage site as determined by the subsequent mass spectroscopy experiments. To this extent the data should not be controversial. We did take further steps to generate a control, reverse mimetic, and from the subsequent experiments we propose (as described above for reviewer 1) that the interaction with the protease is not sequence-specific but rather relies on the general biophysical characteristics, particularly the charge, of the polypeptide.

"6) The "Gel based protein visualization" section in the methods is mostly duplicated and should be consolidated into one section."

We agree with the reviewer and the methods section has been edited and has the following text:

"Gel-based protein visualisation: western blotting

Cell supernatants or lysates (prepared in Cell Lysis Buffer, Cell Signaling) were mixed with NuPAGE LDS Sample Buffer and sample reducing agent (Life Technologies) and run on 4-12% w/v BIS-TRIS gels (Life Technologies). Gels were transferred to PVDF membrane which was probed with AF1828, goat anti-hTREM2 N-terminal domain or MAB1729, rat anti-mTREM2-N-terminal domain (both from R&D Systems, 2 µg/ml in Odyssey block or 5% w/v non-fat milk).

Blots in the siRNA experiments were probed with polyclonal Rabbit anti-ADAM10 (14194S, Cell Signaling), rabbit anti-ADAM17 (AB19027, Millipore) and mouse anti- β -actin (ab3280, Abcam). Immunocomplexes were detected using the following secondary antibodies: rabbit anti-mouse (Thermo Scientific), rabbit, mouse or goat IgG HRP-conjugated Antibody (R&D Systems), IRDye[®] 680RD donkey anti-goat IgG or IRDye[®] 800CW goat anti-rat IgG. Molecular weight ladders used were Chameleon Duo Pre-stained Protein Ladder (LI-COR) or MagicMark[™] XP (ThermoFisher). Blots were either visualised on LI-COR Odyssey imaging system or using photographic film.”

Reviewer 3

We were happy to read the supportive comments from this reviewer.

Reference:

Caescu, C. I., Jeschke, G. R. and Turk, B. E. (2009). Active-site determinants of substrate recognition by the metalloproteinases TACE and ADAM10. *Biochem. J.* **424**, 79–88.

2nd Editorial Decision

18 July 2017

Thank you for the submission of your revised manuscript to EMBO Molecular Medicine. We have now received the enclosed reports from the referee who was asked to re-assess it. As you will see the reviewer is now globally supportive and I am pleased to inform you that we will be able to accept your manuscript pending the following editorial final amendments - I'd like to let you know that the paper from the Haass's lab has been accepted last Friday. Therefore, in order to not delay online publication of both papers, we would encourage you to address the remaining editorial amendments as soon as possible. I also want to let you know that the EMBO Communications team is working on a Press Release on both papers and they will contact you separately soon.

1) Figures:

Western blot provided in fig EV3 is of low resolution, if you could replace it with higher resolution image, that would be great

Western blots from fig 5A are juxtaposed. Could you please either leave a clear blank to indicate they come from different source or use a black line in between.

make sure that your figures can be printed in a portrait format, not landscape or you will be asked by production to modify them.

2) Source Data:

We now encourage the publication of source data, particularly for electrophoretic gels, blots, but also microscopy images with the aim of making primary data more accessible and transparent to the reader. Would you be willing to provide a PDF file per figure that contains the original, uncropped and unprocessed scans of all or key gels used in the figure? The PDF files should be labeled with the appropriate figure/panel number (1 file/figure), and should have molecular weight markers; further annotation may be useful but is not essential. The PDF files will be published online with the article as supplementary "Source Data" files. If you have any questions regarding this just contact me.

3) In the main manuscript file, please add the following:

- in M&M, include a statement that the experiments performed with human samples conformed to the principles set out in the WMA Declaration of Helsinki and the Department of Health and Human Services Belmont Report.
- in M&M, were the neonatal pups bought alive? if so, please confirm that all experiments were performed in accordance with relevant guidelines and regulations. The manuscript must include a statement in the Materials and Methods identifying the institutional and/or licensing committee approving the experiments. If they were bought dead, indicate it.
- in M&M, provide the antibody dilutions that were used for each antibody
- in M&M, provide a statistical paragraph. Please make sure to populate this statistical paragraph according to all the questions asked in the author checklist that you have filled.

Please submit your revised manuscript as soon as possible but within two weeks.

I look forward to it.

***** Reviewer's comments *****

Referee #2 (Remarks):

The authors have addressed my comments in a positive and satisfactory manner.

2nd Revision - authors' response

31 July 2017

Authors made the requested editorial changes.

Corresponding Author Name: Damian Crowther

Manuscript Number: EMM-2017-07673-V2